

# Tackling critical parameters in metazoan meta-barcoding experiments: a preliminary study based on *coxI* DNA barcode

Bachir Balech[1,2], Anna Sandionigi[3], Caterina Manzari[1], Emiliano Trucchi[4], Apollonia Tullo[1,5], Flavio Licciulli[5], Giorgio Grillo[5], Elisabetta Sbisà[5], Stefano De Felici[4,6], Cecilia Saccone[7], Anna Maria D'Erchia[7], Donatella Cesaroni[4], Maurizio Casiraghi[3] and Saverio Vicario[8]

[1] Istituto di Biomembrane, Bioenergetica e Biotecnologie Molecolari–Consiglio Nazionale delle Ricerche, Bari, Italy
[2] Dipartimento di Biologia, Università degli studi di Bari 'Aldo Moro', Bari, Italy
[3] Dipartimento di Biotecnologie e Bioscienze–Zooplantlab, Università degli studi di Milano Bicocca, Milan, Italy
[4] Dipartimento di Biologia, Università di Roma Tor Vergata, Rome, Italy
[5] Istituto di Tecnologie Biomediche–Consiglio Nazionale delle Ricerche, Bari, Italy
[6] Istituto di Biologia Agroambientale e Forestale–Consiglio Nazionale delle Ricerche, Rome, Italy
[7] Dipartimento di Bioscienze, Biotecnologie e Biofarmaceutica, Università degli Studi di Bari 'Aldo Moro', Bari, Italy
[8] Istituto sull'Inquinamento Atmosferico–Consiglio Nazionale delle Ricerche, Bari, Italy

Corresponding author
Bachir Balech,
balechbachir@gmail.com

## ABSTRACT

Nowadays DNA meta-barcoding is a powerful instrument capable of quickly discovering the biodiversity of an environmental sample by integrating the DNA barcoding approach with High Throughput Sequencing technologies. It mainly consists of the parallel reading of informative genomic fragment/s able to discriminate living entities. Although this approach has been widely studied, it still needs optimization in some necessary steps requested in its advanced accomplishment. A fundamental element concerns the standardization of bioinformatic analyses pipelines. The aim of the present study was to underline a number of critical parameters of laboratory material preparation and taxonomic assignment pipelines in DNA meta-barcoding experiments using the cytochrome oxidase subunit-I (*coxI*) barcode region, known as a suitable molecular marker for animal species identification. We compared nine taxonomic assignment pipelines, including a custom in-house method, based on Hidden Markov Models. Moreover, we evaluated the potential influence of universal primers amplification bias in qPCR, as well as the correlation between GC content with taxonomic assignment results. The pipelines were tested on a community of known terrestrial invertebrates collected by pitfall traps from a chestnut forest in Italy. Although the present analysis was not exhaustive and needs additional investigation, our results suggest some potential improvements in laboratory material preparation and the introduction of additional parameters in taxonomic assignment pipelines. These include the correct setup of OTU clustering threshold, the calibration of GC content affecting sequencing quality and taxonomic classification, as well as the evaluation of PCR primers amplification bias on the final biodiversity pattern. Thus, careful

attention and further validation/optimization of the above-mentioned variables would be required in a DNA meta-barcoding experimental routine.

## INTRODUCTION

The introduction of DNA barcoding (*Hebert, Ratnasingham & DeWaard, 2003*) shed new light on the identification process of many life forms on earth leading to a wider comprehension of many ecosystems both aquatic and terrestrial (*Aylagas, Borja & Rodriguez-Ezpeleta, 2014*; *Comtet et al., 2015*). DNA barcoding was originally designed to identify single organisms, but scientific progresses have adapted it to a new but similar technique called "DNA meta-barcoding". This mainly consists of the parallel reading of informative genomic fragment/s able to discriminate living entities (*Taberlet et al., 2012*). DNA barcoding and meta-barcoding share the common process in flagging a specific DNA sequence to a taxonomic name, hopefully at species level (*Hajibabaei, 2012*). However, compared to DNA barcoding, the meta-barcoding approach requires a reduced sampling effort to collectively characterize the inhabitant organisms of a given environment (*Coissac, Riaz & Puillandre, 2012*; *De Barba et al., 2014*). Supported by the High Throughput Sequencing (HTS) technologies (*Bik et al., 2012*; *Shokralla et al., 2012*), DNA meta-barcoding is revolutionizing ecological studies by expanding the information on ecosystem biodiversity (*Kajtoch, 2014*). This innovative tool is also widely used for monitoring purposes such as invasive species control (*Comtet et al., 2015*). Moreover, its ability to identify fungi (*Bellemain et al., 2013*), plants (*Quemere et al., 2013*), chromista (*Nanjappa et al., 2014*), bacteria (*Sogin et al., 2006*) and metazoans (*Leray & Knowlton, 2015*) from the same sample or ecological area is of great importance to understand natural connections among these life forms and consequently to plan ecosystem monitoring and/or biodiversity conservation programs (*Ji et al., 2013*).

Many molecular markers are currently used to characterize species or taxa from all life domains, for instance, ITS (Internal Transcribed Spacer) targets fungi (*Geml et al., 2014*; *Op De Beeck et al., 2014*), 16S ribosomal RNA is employed in bacterial identification (*Fantini et al., 2015*) and *trnL* (UAA) intron (*Srivathsan et al., 2015*) or *rbcL-matK* (large subunit of RUBISCO—Maturase K) (*Xu et al., 2015*) are accustomed to plants. Besides, the use of the cytochrome oxidase subunit-I (*coxI*) molecular marker in animals' identification is increasingly gaining greater importance in DNA barcoding and meta-barcoding studies (i.e., *Mollot et al., 2014*). Thanks to HTS technologies, the above-mentioned molecular markers can be used simultaneously in order to target a broad range of life forms from the same experimental sample (e.g., *Gibson et al., 2014*; *Zhan et al., 2014*). However, this massive and integrative approach is still method sensitive, which requires different levels of standardization and optimization in its implementation steps (*Cristescu, 2014*), including both laboratory materials and techniques and bioinformatics analyses. Once all those steps

are standardized, it becomes possible to compare different experiments carried out by a meta-barcoding approach. For instance, the use of efficient and low-biased universal PCR primer pairs is crucial to get a successful DNA amplification of the majority of organisms present in the environmental sample under investigation. Recent studies highlighted the influence of universal primers amplification bias on the accuracy of species diversity reconstruction and their relative abundances (*Geisen et al., 2015*; *Pawluczyk et al., 2015*; *Pinol et al., 2015*). Sequencing chemistries prone to errors caused by GC content are also of fundamental importance as they can alter the true picture of organismal composition (*Abnizova et al., 2012*). Additional variables, which may influence the quality of meta-barcoding sequences processing, include the availability of well-represented reference databases (*Cowart et al., 2015*) and the compatibility of taxonomic assignment tools with the DNA information of the molecular marker in use (*Balint et al., 2014*). Consequently, to reach the relevant conclusions from a meta-barcoding assay, bioinformatic analyses pipelines would require some standardization taking into account the aforementioned variables. A typical pipeline in DNA meta-barcoding data analysis consists of three parts: (1) sequence denoising or quality filtering; (2) operational taxonomic units (OTU) picking and (3) taxonomic assignment at species and/or higher levels. Once the denoising protocol has been defined (*Edgar et al., 2011*; *Ficetola et al., 2015*; *Quince et al., 2011*; *Schloss, Gevers & Westcott, 2011*), two critical and dependent parameters can potentially influence the quality of taxonomic assignment, namely OTU clustering (*Chen et al., 2013*; *Edgar, 2013*) and classification (*Bacci et al., 2015*) thresholds. The clustering threshold is directly correlated to the intra-specific distance and at the same time to the sequencing error rate. This can guide significantly the accuracy and precision of the subsequent taxonomic identification, influenced by the correct classification threshold, and the relative abundance assigned to each classified taxon.

Here we present a preliminary study aimed to highlight a set of critical parameters of taxonomic assignment pipelines used in *coxI* DNA meta-barcoding analyses, namely OTU clustering, GC content and PCR primers amplification bias. For that, we compared nine taxonomic assignment pipelines, of which a custom in-house one based on Hidden Markov Models (HMM). The taxonomic assignments obtained by the best pipeline were then evaluated for their potential influence by universal primers amplification bias (using qPCR) and by GC content. The pipelines were tested on a community of known terrestrial invertebrates, taxonomically classified based on their morphology, collected by pitfall traps from a chestnut forest in Italy.

## MATERIALS AND METHODS

### Samples description

The sampling unit consisted of two samples containing soil litter macrofauna, called MPE4 and MPE5. MPE5 is the result of pooling the content of five pitfall traps, placed in circle at 10 m of distance each, in a chestnut forest located in central Italy (province of Rome). All collected organisms belonging to the *Carabidae* family (order: *Coleoptera*) were morphologically classified at species level, while the others were labeled with their order
**Table 1 Taxonomically identified organisms in MPE5 sample and their corresponding biomass.** MPE4 simply contains the same organisms at equal biomass.

| Taxonomic group | Total biomass (g) | Species name | Biomass of single organism (g) |
|---|---|---|---|
| Coleoptera (family: Carabidae) | 22.89 | Carabus (Tomocarabus) convexus dilatatus[b] | 12.94 |
| | | Carabus (Chaetocarabus) lefebvrei bayardi | 4.85 |
| | | Calathus fracasii | 0.58 |
| | | Abax parallelepipedus | 0.29 |
| | | Laemostenus latialis[b] | 0.23 |
| | | Pterostichus micans | 0.18 |
| | | Calathus montivagus | 0.07 |
| Diptera | 4.12 | | |
| Orthoptera | 2.44 | | |
| Blattodea | 1.02 | | |
| Myriapoda[a] | 0.82 | | |
| Isopoda | 0.7 | | |
| Arachnida[a] | 0.64 | | |
| Scorpiones | 0.63 | | |
| Hymenoptera | 0.21 | | |
| Lepidoptera | 0.1 | | |
| Collembola | 0.02 | | |

**Notes.**
[a] Class taxonomy rank.
[b] Laemostenus latialis = LL1 and Carabus (Tomocarabus) convexus dilatatus = CC1

(whenever possible) or class rank names. The biomass of each organism was measured singularly for further statistical analysis. MPE4 is an *ad-hoc* sample composed from the equal biomass content of all the organisms present in MPE5, using the same weight of the lightest organism (see Table 1 for organisms' names and their corresponding biomass).

## DNA extraction and amplification of *coxI* barcode

Each sample, MPE4 and MPE5, was homogenized separately and total genomic DNA was extracted using the DNeasy Blood and Tissue (Qiagen, Hilden, Germany) commercial kit. In addition, total DNA was extracted from the *Carabidae* species individually. The *coxI* DNA barcode was amplified from all DNA extracts using the universal primer pairs (*Folmer et al., 1994*): forward-LCO1490 (5′-GGTCAACAAATCATAAAGATATTGG-3′), and reverse-HCO2198 (5′-TAAACTTCAGGGTGACCAAAAAATCA-3′). PCR reactions were carried out in 50 μl reaction volumes containing: 1.5 mM MgCl2, 250 nM of each primer, 200 μM of each dNTP, 1x of Phusion HF Buffer, 1U of Phusion DNA polymerase (M0530S, NEB) and 2 μl of DNA extracts, using a thermocycling profile of one cycle of 60 s at 94 °C, five cycles of 60 s at 94 °C, 90 s at 45 °C, and 90 s at 72 °C, followed by 35 cycles of 60 s at 94 °C, 90 s at 50 °C, and 60 s at 72 °C, with a final step of 5 min at 72 °C. PCR products along with 100 bp DNA Ladder (Fermentas Life Sciences, Waltham, MA, USA) were visualized on a 1% agarose gel stained with 0.005% of ethidium bromide.

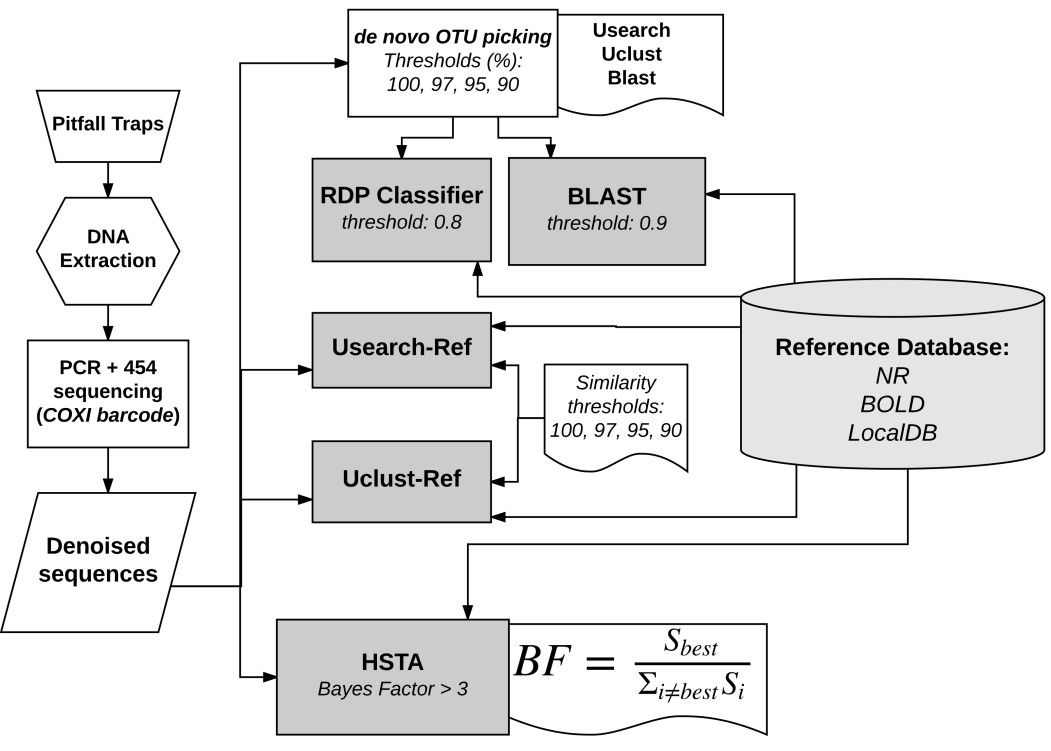

**Figure 1 Bioinformatics analysis pipelines overview.** The same reference database is used by all methods to assign each sequence to a known taxon name. RDP and BLAST classifiers take as input the output of OTU picking methods (Usearch, Uclust, Blast). HSTA, Usearch-Ref and Uclust-Ref use the denoised sequences as direct input. The Bayes Factor (*BF*) equation is a logical representation of that implemented in HSTA (see OTU picking and Taxonomic Assignment section, letter c.)

PCR products were subsequently gel purified using QIAquick Gel Extraction Kit (Qiagen, Hilden, Germany).

## Sequencing

The *coxI* barcode region was sequenced singularly for each *Carabidae* organism by Sanger method. These sequences were then used as controls in the following experimental and analysis steps. Furthermore, MPE4 and MPE5 samples were prepared for pyrosequencing adapting the sequencing library preparation protocol described by *Calabrese et al. (2013)*. The libraries were then deposited on 2/8-lane PicoTiterPlate (PTP) wells (Roche/454; Roche, Basel, Switzerland) and sequenced in both directions on GS FLX Titanium pyrosequencing platform.

## Bioinformatics analysis

The bioinformatics analysis was mainly divided into three steps: (i) sequence reads filtering and denoising, (ii) OTU picking and (iii) taxonomic assignment. In this study, we compared nine pipelines (Fig. 1), of which a custom one based on Hidden Markov Models. For the sake of simplicity, in the rest of the manuscript we will call the custom pipeline HSTA (Hidden States Taxa Assign). However, we followed a standard procedure in the first

step and used several OTU picking algorithms with progressive thresholds and different taxonomic assigners in the second and third steps respectively. The tested pipelines are based on (1) Usearch, Uclust and Blast as OTU picking methods (thresholds: 100, 97, 95, 90%) in combination with RDP and BLAST classifiers (all available in Qiime *Caporaso et al., 2010*), (2) Usearch-Ref, Uclust-Ref (thresholds: 100, 97, 95, 90%) (*Edgar, 2013*) and HSTA as direct taxonomic assigners without calling OTUs.

## Sequence reads filtering and denoising

We performed a standard read rejecting and trimming procedure, using GS Run Processor V2.4 (Roche 454 Life Sciences software package; Roche, Basel, Switzerland), with the parameters suggested by the 454 platform manufacturer. Due to amplicons ligation and subsequent nebulization steps carried out for sequencing library preparation (see *Calabrese et al., 2013*), we executed a primer search analysis (Supplemental Information 1) generating forward and reverse reads data sets, which were processed separately in the subsequent analyses. Pyrosequencing and PCR noises were removed by invoking PyroNoise and SeqNoise algorithms respectively, available from AmpliconNoise package (*Quince et al., 2011*), with their default parameters. Denoised sequences were then submitted to chimera detection and removal using UCHIME (de novo method) (*Edgar et al., 2011*). At this point, we obtained two denoised data sets per sample corresponding to the 5′ and 3′ ends of *coxI* barcode region.

## OTU picking and taxonomic assignment

We applied the *de-novo* OTU picking method of Usearch, Uclust and Blast algorithms, available in Qiime (*Caporaso et al., 2010*), at four different similarity thresholds: 100, 97, 95 and 90%.

OTUs were taxonomically classified by RDP and BLAST classifiers with their default parameters. In addition, without calling OTUs, a direct taxonomic assignment was carried out by Usearch-Ref and Uclust-Ref (*Edgar, 2013*) at four different similarity thresholds (100, 97, 95 and 90%) and by HSTA. Note that the same *coxI* reference database was used for all tested pipelines (see below for additional information). Given that HSTA was assembled as a custom pipeline, in the following we provide a description of its main steps, namely (a) the choice of reference sequences, (b) reference Hidden Markov Models (HMM) profiles building and (c) taxonomic classification method.

### (a) Choice of reference sequences

Several sources of *coxI* reference sequences were taken into account in building a *coxI* reference database. First of all, a local database (LocalDB) was generated from the *coxI* DNA barcode sequences, produced in this study, of 114 *Carabidae* individuals belonging to seven different species classified morphologically at species level (species names are listed in Table 1). On the other hand, public *coxI* sequences were obtained by blastx (*Altschul et al., 1990*) using the denoised sequences as queries against BOLD (http://boldsystems.org/) and GenBank (NR-NCBI) databases. Blastx outputs were parsed by Biopython1.57 script and the first 10 best matches' IDs per query were retrieved and tagged in their order rank name

(NCBI taxonomy). All 10 matched references per query were filtered to select only the entries which had the same taxonomical order of the best-hit match.

### (b) Reference HMM profiles building

Amino acid reference sequences selected in the previous step (belonging to each order and species ranks) were multiple aligned using Muscle 3.8.31 (*Edgar, 2004*). Multiple protein alignments were used to generate the relevant nucleotide ones, denoted as back-alignment, as described by *Balech et al. (2015)*. Hidden Markov Model (HMM) profiles were then built from the nucleotide alignments using *hmmbuild* (HMMer 3.0) with its default parameters (*Finn, Clements & Eddy, 2011*).

### (c) Taxonomic classification method

Denoised sequences were assigned to one of the nucleotide HMM profiles by executing *hmmscan* (HMMer 3.0) with its default parameters. The outputs were then parsed using a Python 2.7 script which classifies the assigned sequence in three categories:

– Unclassified assignment: sequence-profile match outputs an e-value higher than *hmmscan* default one.
– Good assignment: the best match bit score passes the threshold of Bayes Factor (*BF*) (Eq. (1)), set to 3.0. *BF* is computed by subtracting the best bit score ($S_{i=1}$) from the natural logarithm of the sum of all ($S_{i=2-n}$) remaining exponential bit scores.

$$\ln(BF) = S_{i=1} - \ln(\sum_{i=2}^{n} \exp(S_i)) \tag{1}$$

– Ambiguous assignment: sequence-profile hit passes *hmmscan* threshold but does not pass *BF* test.

## Taxonomic assignment pipelines comparison

To validate how well the tested pipelines fit, we calculated the ratio of the detected taxa by each tested pipeline against the expected ones. Thus, the taxonomically classified reads were searched for the expected *Carabidae* species and order level taxa listed in Table 1. In addition, Pearson's correlations were calculated between read abundances relative to each taxon and its corresponding biomass.

## GC content assessment

GC content assessment was conducted to infer its potential relationship with sequencing errors and consequently its influence on the denoising and taxonomic assignment processes behavior. We therefore hypothesized that the denoising procedure should be calibrated based on GC content effect, which potentially influences the sequencing reaction. To test this hypothesis, we assessed the contribution of the expected biological variation and the GC content (both present in the reference sequences) on denoised reads variation taking into consideration the sequences assigned to *Coleoptera* order (the taxonomic group with most abundant assigned reads and reference sequences). We used a linear model "Eq. (2)" to predict the change in diversity across sites.

$$Dreads \sim Dref + GCref \tag{2}$$

In (Eq. (2)) *Dreads* and *Dref* are the exponential entropy per site in reads assigned to *Coleoptera* and the reference sequences respectively, while *GCref* is the mean GC percentage over a 100bp window in the reference sequences. We used the exponential entropy per site following *Jost (2006)* as an example of the diversity index for categorical variable. The model was estimated separately for the 5′ *coxI* barcode, taking the first 400 bp of the *Coleoptera* nucleotide multiple alignments (reference and assigned reads), and the 3′ one starting from the 200th site until the end.

## Species level assignment evaluation with qPCR

To explain the correlation observed between the abundance of assigned sequences and taxon biomass at species level, we adopted a qPCR approach to track the number of *coxI* copies in two different *Carabidae* species across all experimental steps. We therefore compared the ratio of *coxI* copies, the biomass and the assigned sequence abundance obtained from HSTA (the pipeline that detected the highest number of known species). Accordingly, two specific primer pairs were designed for CC1 (*Carabus* (*Tomocarabus*) *convexus dilatatus*; forward: 5′-ATTCTGGCTCCTACCTCCG-3′, reverse: 5′-CTGCCCCTAAAATTGATGAG-3) and LL1 (*Laemostenus latialis*; forward: 5′-TACGATCTACAGGAATAACC-3′, reverse: 5′-AGCAGGGTCAAAAAAGGAT-3′). qPCR reactions, using SYBR Green as a fluorescent reporter, were conducted in 50 μl reaction volumes containing 2 μl of DNA template, 300 nM of each primer and 22.5 μl of RealMasterMix SYBR ROX 2.5× (5 Prime): 1U Taq DNA Polymerase, 4 mM Magnesium Acetate, 0.4 mM of each dNTP, using a thermocycling profile of one cycle of 2 min at 94 °C, 45 cycles of 30 sec at 94 °C, 30 sec at 60 °C and 40 sec at 68 °C. The DNA templates consisted of MPE4 and MPE5 DNA extracts, their PCR products obtained from the amplification with Folmer primer set and the DNA yielded from libraries preparation for 454 sequencing. The reactions were performed on a 7900 HT Fast Real-time PCR system (Applied Biosystems, Foster City, CA, USA). A total of 18 replicates per primer set and per template were performed (six replicates per template over three different qPCR plates) and the resulting data was analyzed by MAK2 (*Boggy & Woolf, 2010*).

## RESULTS

### Sequencing yield and taxonomic assignment

A total of 160,166 passed filter reads were obtained for both samples (MPE4: 67672, MPE5: 92494). Following the primer search analysis (Supplemental Information 1) and subsequent denoising, 2595 denoised reads were obtained for MPE5 (650 for the 5′ and 1945 for the 3′) and 5368 for MPE4 (1271 for the 5′ and 4097 for the 3′). In HSTA and reference-based assignments (Usearch-Ref and Uclust-Ref), we used denoised reads as a direct input for down streaming analysis, while we applied an OTU picking step for the other methods (see materials and methods).

Comparing the taxonomic assignment results obtained in at order level, at 5′ end (Fig. 2A) HSTA could detect seven out of 11 expected orders followed by five orders identified by Usearch+BLAST (OTU picking method+Classifier) at 97, 95 and 90% and Uclust+BLAST at 95 and 90% OTU picking thresholds. As for the 3′, five orders were

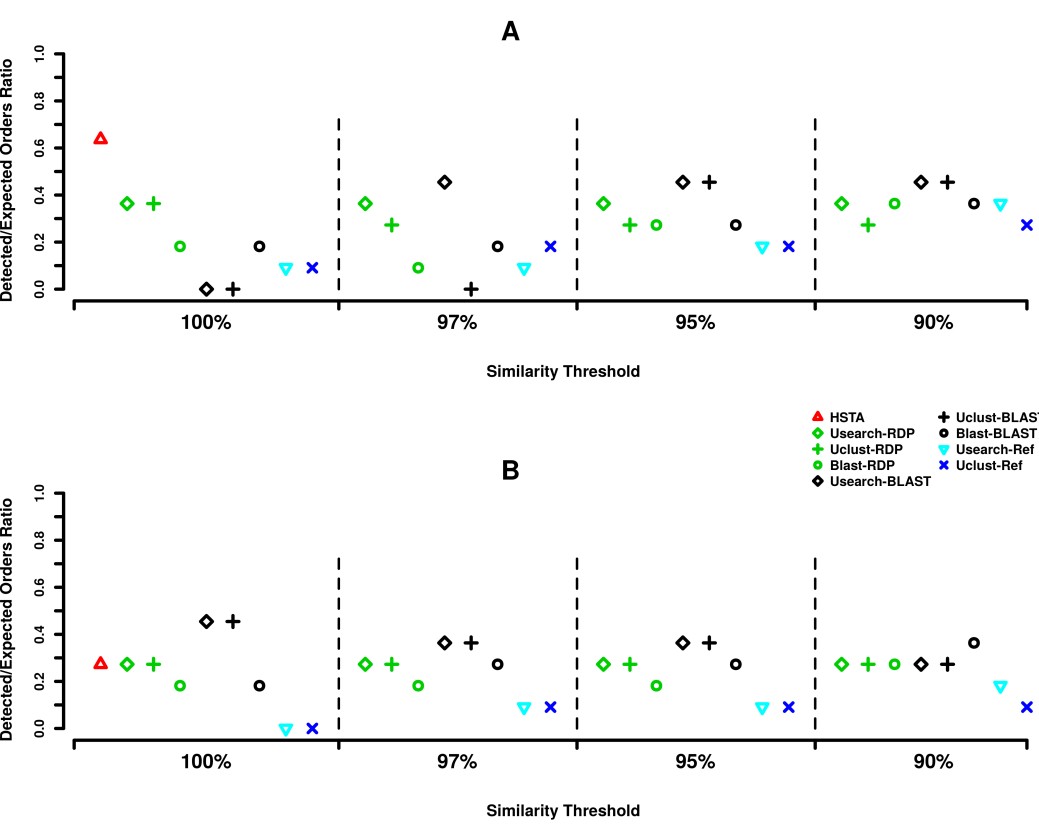

**Figure 2** **Order level taxonomic assignments in MPE5 sample.** (A) Forward strand or 5′ *coxI*, (B) reverse strand or 3′ *coxI*. Each color corresponds to a taxonomic classifier (HSTA, RDP, BLAST) or a reference-based assigner algorithm (Usearch-Ref, Uclust-Ref). The symbols indicate either the combination of an OTU picking method with a classifier or the classifier/algorithm used for direct taxonomic assignment. The *x*-axis corresponds to the similarity thresholds used in OTU picking or with the direct assignments in HSTA and reference-based algorithms.

found by Usearch+BLAST and Uclust+BLAST at 100% OTU picking threshold followed by HSTA detecting only three (Fig. 2B).

Regarding the assignment at species level (Fig. 3), out of seven expected species, HSTA could classify all of the searched species in both the 5′ and 3′ datasets. Furthermore, at 5′, Blast+RDP (OTU picking method+Classifier) showed stable behavior as it detected five species at all OTU picking thresholds. The same result was obtained for Usearch+BLAST at 97 and 90% and Uclust+BLAST at 95 and 90% OTU picking threshold (Fig. 3A). At the 3′ end (Fig. 3B), the OTU picking methods Uclust and Blast with RDP classifier appear to be consistent over all similarity thresholds classifying four out of seven expected species. Moreover, it is important to note that reference-based algorithms (Usearch-Ref and Uclust-Ref) were able to uncover a lower number of expected taxa than classifiers did.

Pearson's correlation values (calculated only when more than one taxon was classified) of the assigned reads abundance against the corresponding taxa biomass demonstrated linear behavior with the qualitative results described above, confirming the accuracy of almost all tested taxonomic assignment pipelines. Overall, these correlations were higher at order

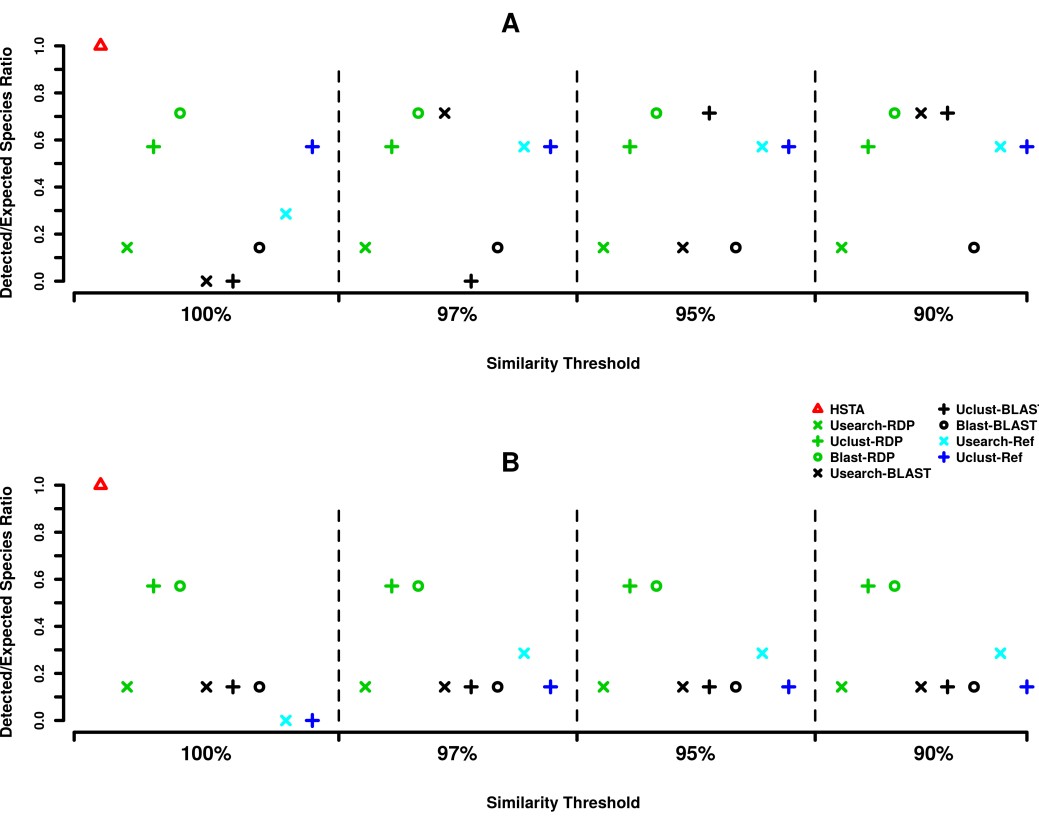

**Figure 3** **MPE5 taxonomic assignments plots at species level.** (A) Forward strand or 5′ *coxI*, (B) reverse strand or 3′ *coxI*. Each color corresponds to a taxonomic classifier (HSTA, RDP, BLAST) or a reference-based assigner algorithm (Usearch-Ref, Uclust-Ref). The symbols indicate either the combination of an OTU picking method with a classifier or the classifier/algorithm used for direct taxonomic assignment. The *x*-axis corresponds to the similarity thresholds used in OTU picking or with the direct assignment in HSTA and reference-based algorithms.

level assignments (greater or equal to 0.98) compared to species ones (ranged from 0.86 to 0.92). The only two exceptions, at species level, showing respectively negative correlations of −0.22 and −0.79 at 100% similarity threshold were Blast+RDP and Uclust-Ref.

## GC content assessment

GC content assessment was conducted separately for the 5′ *coxI* barcode, taking the first 400bp of the *Coleoptera* nucleotide multiple alignment, and the 3′ one starting from the 200th site until the end. As expected, the linear model results (Table 2) showed that the diversity index (exponential value of entropy) of the assigned sequences is explained by that of reference sequences. In addition, the variability of diversity value in denoised sequences weighed by GC content was statistically significant at both the 5′ and 3′ *coxI* barcode. Nevertheless, the sum of squares values pointed out that this significant variability due to GC content is larger at the 3′ (38 corresponding to 17% of explained variance) than that at the 5′ (4.5 equivalent to 2.4%). This indicates some shortcomings in denoising protocol given by its potential inability to remove the excess of errors due to GC content.

**Table 2  Sites diversity in sequence reads predicted by sites diversity in references and by GC content.** The results are reported for sequences assigned to *Coleoptera* order over the 5′ and 3′ end of *coxI* barcode region.

| | 5′*coxI* barcode analyzed sites: 1-400 | | | 3′*coxI* barcode analyzed sites: 200-end | | |
|---|---|---|---|---|---|---|
| | GCref | Dref | Residuals | GCref | Dref | Residuals |
| Degree of freedom | 1 | 1 | 562 | 1 | 1 | 562 |
| Sum of squares | 4.518 | 58.766 | 123.808 | 37.999 | 50.025 | 135.407 |
| Mean square | 4.518 | 58.766 | 0.220 | 37.999 | 50.025 | 0.241 |
| *F*-value | 20.507 | 266.754 | – | 157.71 | 207.63 | – |
| Explained variance (%) | **2,41** | 31,41 | 66,17 | **17** | 22,3 | 66,17 |
| Pr($>$ F) | **7.257e−06**\*\*\* | <2.2e−16\*\*\* | – | **<2.2e−16**\*\*\* | <2.2e−16\*\*\* | – |

Notes.

*GCref* and *Dref* are the GC content and sites diversity of reference sequences respectively.

Significance codes: \*\*\*, 0; \*\*, 0.001; \*, 0.01; ., 0.05.

**Table 3  Universal primers amplification bias analysis results of MPE5 and MPE4 samples.** All the reported values refer to the ratio CC1/LL1 of qPCR intensity signal for DNA extract, PCR products and library preparation categories while 5′and 3′counts correspond to the number of assigned reads for CC1 and LL1 species obtained from HSTA pipeline. LL1 = *Laemostenus latialis*, CC1 = *Carabus (Tomocarabus) convexus dilatatus*.

| | | qPCR signal intensity | | | HSTA assigned reads | |
|---|---|---|---|---|---|---|
| Sample | Biomass | DNA extract | PCR products | Library preparation | Count at 5′ | Count at 3′ |
| MPE5 | 56.26 | 1508.46 | 620.31 | 122.34 | 231.35 | 103.55 |
| MPE4 | 1 | 53.92 | 163.76 | 19.59 | 1.83 | 1.29 |

## Species level assignment validation with qPCR

To perform a quantitative validation of organisms biomass effect on taxonomic assignments, we chose two organisms, CC1 and LL1 (Table 1), belonging to the *Carabidae* family and classified morphologically at species level. We calculated the ratio CC1/LL1 of biomass and that of read abundance obtained from HSTA (the pipeline that detected the highest number of known species) (Table 3) in both MPE4 and MPE5 samples. At equal biomass (MPE4) CC1 had almost an average of 1.6 (=(1.83 at 5′ + 1.29 at 3′)/2) greater assigned reads than LL1, while at 53 biomass ratio (in MPE5) read abundance increased significantly to approximately 167 times (=(231.35 at 5′ + 103.55 at 3′)/2) of CC1 over LL1. Moreover, the ratio recorded in DNA extracts appears to be altered when compared to that of PCR products, as it decreased in MPE5 (approximately 620) and increased in MPE4 (approximately 164). This emphasizes a potential amplification bias in both samples that is however more manageable in MPE4, where the difference between the initial biomass content and the final read abundance ratios is almost negligible.

## DISCUSSION

Taxonomic profiling based on HTS technologies of species discriminant loci has been and is still being widely used in numerous microbial and invertebrates biodiversity studies where it has showed its feasibility and accuracy in monitoring species of ecological and/or clinical relevance (*Brulc et al., 2009*; *Cristescu, 2014*; *Fonseca et al., 2014*; *Fonseca et al., 2010*;

*Singh et al., 2009*). Considering two control samples of known metazoan organisms, we addressed some of the parameters potentially influencing the outcomes of a *coxI* meta-barcoding experiment. It can be argued that we did not perform a massive validation on a wide range of samples and on other molecular barcode markers except *coxI*. However, as a preliminary study the taxonomic assignment pipelines showed promising results in classifying the known organisms already taxonomically classified based on their morphology and their *coxI* barcode region (sequenced singularly by Sanger method). We therefore reported, according to this specific experiment, some areas of improvement in both laboratory materials preparation and taxonomic assignment methods.

Based on our taxonomic assignments results, the use of a classifier appear to be essential in *coxI* meta-barcoding as the direct call of the known organisms by reference-aware algorithms (i.e., Userach-Ref and Uclust-Ref) could not satisfy the same outcome as classifiers did. Combining the results observed for the assignment at order level (Fig. 2), the use of OTU picking methods, Usearch and Uclust, with BLAST classifier showed consistent and similar behavior at both the 5′ and 3′ ends (0.45% of detected orders). However, the best results of these pipelines were obtained using different OTU clustering thresholds (97, 95, 90% at 5′ and 100% at 3′). This underlines the importance of this parameter in taxonomic assignment routine and suggests its calibration according to an internal control sample or a mock community with similar taxonomic composition of the true samples.

Considering the assignment at species level (Fig. 3), the HSTA assigner, based on Hidden Markov Models, showed promising results as it could detect the highest number of the searched known organisms at both the 5′ and 3′ ends. Although this pipeline is only at a prototype status and needs additional validations and improvements, its strength resides in the absence of an OTU clustering step and in the possibility to extend it to a phylogenetic assignment framework (e.g., pplacer (*Matsen, Kodner & Armbrust, 2010*), RaxML (*Stamatakis, 2014*)), as HMM profiles are built from multiple sequence alignments. Alternatively, it is important to mention that Blast+RDP (OTU picking method+Classifier) also showed stable classification behavior at both *coxI* ends as they detected five species (0.71% of the expected species) independently from OTU picking thresholds. The differences in accuracy among the tested pipelines while varying the taxonomic level (order or species) would be due to the characteristics of *coxI* as molecular marker. It is worth mentioning that the majority of the tested methods were specifically designed to analyze 16S rRNA meta-barcoding data (*Caporaso et al., 2010*). Similar observations were also reported by *Balint et al. (2014)* for the Internal Transcribed Spacer (ITS) analysis pipelines, where the correct taxonomic assignment was influenced by the used tools.

Concerning GC content, we investigated its effect within the *Coleoptera* reference data set, being the biggest data set used in this study and at the same time containing the organisms present in our local reference database (see materials and methods OTU picking and Taxonomic Assignment section). GC content is known to influence the quality of sequence reads in both pyrosequencing (*Hoff, 2009*) and Illumina sequencing (*Abnizova et al., 2012*). In the present work, we clarified this impact on taxonomic assignment pipelines even after applying a standard denoising protocol. In fact, the low correlation between biomass content and the corresponding assigned reads abundance per taxon at species level

and the failure to detect several orders, mainly at the 3′, in all tested pipelines would be related to GC content. Indeed, the linear regression model results (Table 2) showed that the variability of assigned reads, higher at 3′, was explained by GC content effect. These results highlights the importance of this parameter and proposes its introduction in denoising or quality filtering algorithms of meta-barcoding data to improve taxonomic assignment accuracy.

Another important aspect not to be ignored in similar environmental DNA sequencing (i.e., eDNA) is the amplification bias of universal primer set used in PCR preparation step. Many studies have illustrated the effect of amplification bias and primers conservation on sequencing yield and therefore on the assigned taxa abundances (*Deagle et al., 2014*; *Op De Beeck et al., 2014*; *Pawluczyk et al., 2015*), as well as the influence of biomass on the correct biodiversity estimation through DNA meta-barcoding experiments (*Thomas et al., 2015*). Our investigation in this context using a community of known invertebrates is in line with what is stated above. qPCR results demonstrated a considerable fluctuation of species quantity ratio across all experimental steps in MPE5 sample. The same analysis performed on MPE4, a biomass equalized organismal content, provided a more conserved quantity ratios of those two species (Table 3). This emphasizes a potential limit related to the quantitative prospective of *coxI* DNA meta-barcoding in natural samples (*Pinol et al., 2015*) and suggests, whenever possible, sample manipulation including biomass equilibration prior to sequencing.

In conclusion, we observe a rapid and ongoing increase in the use of *coxI* meta-barcoding assays to study molecular biodiversity. However, the taxonomic assignment pipelines remains challenging. In this preliminary study, we highlighted some important parameters, namely the OTU clustering threshold, GC content and PCR primers amplification bias, to be considered in both laboratory protocols and down streaming analyses. We therefore suggest their extensive evaluation and eventually their introduction into *in-silico* and *in-vitro* data processing routines. This could be achieved by introducing an internal control sample or a mock community in a sequencing run in order to tackle the above-mentioned parameters and by testing novel assignment methods (i.e., HSTA) on a wide range of samples and on different molecular markers.

## ACKNOWLEDGEMENTS

We dedicate this article to Cecilia Lanave, long term partner in research of CS, whose coordinated the effort in writing this project. Unfortunately, an abrupt illness did not allow her to see the end of the project. We would like also to thank Giacinto Donvito, Pasquale Notarangelo and Gaetano Scioscia for providing the access to computational facilities to run AmpliconNoise, Blast and HMM analyses. The authors are also in debt to Michela Barbuto, Annamaria Paluscio and Angela Fortunato for their precious support in sampling and molecular laboratory analyses. Finally, we thank Catriona Isobel Macleod for English revision.

### Funding

This study was funded by 'MIUR - PRIN 2007' project. The funders had no role in study design, data collection and analysis, decision to publish, or preparation of the manuscript.

### Grant Disclosures

The following grant information was disclosed by the authors:
'MIUR - PRIN 2007' project.

### Competing Interests

The authors declare there are no competing interests.

### Author Contributions

- Bachir Balech conceived and designed the experiments, analyzed the data, contributed reagents/materials/analysis tools, prepared figures and/or tables, authored or reviewed drafts of the paper, approved the final draft.
- Anna Sandionigi and Giorgio Grillo analyzed the data, authored or reviewed drafts of the paper, approved the final draft.
- Caterina Manzari, Emiliano Trucchi and Stefano De Felici performed the experiments, authored or reviewed drafts of the paper, approved the final draft.
- Apollonia Tullo performed the experiments, contributed reagents/materials/analysis tools, authored or reviewed drafts of the paper, approved the final draft.
- Flavio Licciulli analyzed the data, contributed reagents/materials/analysis tools, authored or reviewed drafts of the paper, approved the final draft.
- Elisabetta Sbisà, Cecilia Saccone and Anna Maria D'Erchia contributed reagents/materials/analysis tools, authored or reviewed drafts of the paper, approved the final draft.
- Donatella Cesaroni authored or reviewed drafts of the paper, approved the final draft.
- Maurizio Casiraghi conceived and designed the experiments, authored or reviewed drafts of the paper, approved the final draft.
- Saverio Vicario conceived and designed the experiments, analyzed the data, authored or reviewed drafts of the paper, approved the final draft.

### Data Availability

Balech, Bachir (2017): RawSequences_454.zip. figshare. Dataset;

Balech, Bachir (2017): DenoidedSequences.zip. figshare. Dataset;

Balech, Bachir (2017): ColeopteraAndLocalDatabases.zip. figshare. Dataset;

Balech, Bachir (2017): HmmerParser_HSTA.py. figshare. Code;

Balech, Bachir (2017): OrdersProfiles.zip. figshare. Paper;

Balech, Bachir (2017): run_HSTA.sh. figshare. Code;

Balech, Bachir (2017): SpeciesProfiles.zip. figshare. Paper. https://figshare.com/projects/Hidden_States_Taxa_Assign/23782.

## Supplemental Information

Supplemental information for this article can be found online at http://dx.doi.org/10.7717/peerj.4845#supplemental-information.

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

# PeerJ

verifiable and efficient monitoring of biodiversity via metabarcoding. *Ecology Letters* **16**:1245–1257 DOI 10.1111/ele.12162.

**Jost L. 2006.** Entropy and diversity. *Oikos* **113**:363–375 DOI 10.1111/j.2006.0030-1299.14714.x.

**Kajtoch L. 2014.** A DNA metabarcoding study of a polyphagous beetle dietary diversity: the utility of barcodes and sequencing techniques. *Folia Biologica* **62**:223–234 DOI 10.3409/fb62_3.223.

**Leray M, Knowlton N. 2015.** DNA barcoding and metabarcoding of standardized samples reveal patterns of marine benthic diversity. *Proceedings of the National Academy of Sciences of the United States of America* **112**:2076–2081 DOI 10.1073/pnas.1424997112.

**Matsen FA, Kodner RB, Armbrust EV. 2010.** pplacer: linear time maximum-likelihood and Bayesian phylogenetic placement of sequences onto a fixed reference tree. *BMC Bioinformatics* **11**:Article 538 DOI 10.1186/1471-2105-11-538.

**Mollot G, Duyck PF, Lefeuvre P, Lescourret F, Martin JF, Piry S, Canard E, Tixier P. 2014.** Cover cropping alters the diet of arthropods in a banana plantation: a metabarcoding approach. *PLOS ONE* **9**:e93740 DOI 10.1371/journal.pone.0093740.

**Nanjappa D, Audic S, Romac S, Kooistra WH, Zingone A. 2014.** Assessment of species diversity and distribution of an ancient diatom lineage using a DNA metabarcoding approach. *PLOS ONE* **9**:e103810 DOI 10.1371/journal.pone.0103810.

**Op De Beeck M, Lievens B, Busschaert P, Declerck S, Vangronsveld J, Colpaert JV. 2014.** Comparison and validation of some ITS primer pairs useful for fungal metabarcoding studies. *PLOS ONE* **9**:e97629 DOI 10.1371/journal.pone.0097629.

**Pawluczyk M, Weiss J, Links MG, Egana Aranguren M, Wilkinson MD, Egea-Cortines M. 2015.** Quantitative evaluation of bias in PCR amplification and next-generation sequencing derived from metabarcoding samples. *Analytical and Bioanalytical Chemistry* **407**:1841–1848 DOI 10.1007/s00216-014-8435-y.

**Pinol J, Mir G, Gomez-Polo P, Agusti N. 2015.** Universal and blocking primer mismatches limit the use of high-throughput DNA sequencing for the quantitative metabarcoding of arthropods. *Molecular Ecology Resources* **15(4)**:819–830 DOI 10.1111/1755-0998.12355.

**Quemere E, Hibert F, Miquel C, Lhuillier E, Rasolondraibe E, Champeau J, Rabarivola C, Nusbaumer L, Chatelain C, Gautier L, Ranirison P, Crouau-Roy B, Taberlet P, Chikhi L. 2013.** A DNA metabarcoding study of a primate dietary diversity and plasticity across its entire fragmented range. *PLOS ONE* **8**:e58971 DOI 10.1371/journal.pone.0058971.

**Quince C, Lanzen A, Davenport RJ, Turnbaugh PJ. 2011.** Removing noise from pyrosequenced amplicons. *BMC Bioinformatics* **12**:38 DOI 10.1186/1471-2105-12-38.

**Schloss PD, Gevers D, Westcott SL. 2011.** Reducing the effects of PCR amplification and sequencing artifacts on 16S rRNA-based studies. *PLOS ONE* **6**:e27310 DOI 10.1371/journal.pone.0027310.

**Shokralla S, Spall JL, Gibson JF, Hajibabaei M. 2012.** Next-generation sequencing technologies for environmental DNA research. *Molecular Ecology* **21**:1794–1805 DOI 10.1111/j.1365-294X.2012.05538.x.

**Singh J, Behal A, Singla N, Joshi A, Birbian N, Singh S, Bali V, Batra N. 2009.** Metagenomics: concept, methodology, ecological inference and recent advances. *Biotechnology Journal* **4**:480–494 DOI 10.1002/biot.200800201.

**Sogin ML, Morrison HG, Huber JA, Mark Welch D, Huse SM, Neal PR, Arrieta JM, Herndl GJ. 2006.** Microbial diversity in the deep sea and the underexplored "rare biosphere". *Proceedings of the National Academy of Sciences of the United States of America* **103**:12115–12120 DOI 10.1073/pnas.0605127103.

**Srivathsan A, Sha JC, Vogler AP, Meier R. 2015.** Comparing the effectiveness of metagenomics and metabarcoding for diet analysis of a leaf-feeding monkey (Pygathrix nemaeus). *Molecular Ecology Resources* **15**:250–261 DOI 10.1111/1755-0998.12302.

**Stamatakis A. 2014.** RAxML version 8: a tool for phylogenetic analysis and post-analysis of large phylogenies. *Bioinformatics* **30**:1312–1313 DOI 10.1093/bioinformatics/btu033.

**Taberlet P, Coissac E, Pompanon F, Brochmann C, Willerslev E. 2012.** Towards next-generation biodiversity assessment using DNA metabarcoding. *Molecular Ecology* **21**:2045–2050 DOI 10.1111/j.1365-294X.2012.05470.x.

**Thomas AC, Deagle BE, Eveson JP, Harsch CH, Trites AW. 2015.** Quantitative DNA metabarcoding: improved estimates of species proportional biomass using correction factors derivedfrom control material. *Molecular Ecology Resources* **16(3)**:714–726 DOI 10.1111/1755-0998.12490.

**Xu C, Dong WP, Shi S, Cheng T, Li CH, Liu YL, Wu P, Wu HK, Gao P, Zhou SL. 2015.** Accelerating plant DNA barcode reference library construction using herbarium specimens: improved experimental techniques. *Molecular Ecology Resources* **15**:1366–1374 DOI 10.1111/1755-0998.12413.

**Zhan A, Bailey SA, Heath DD, Macisaac HJ. 2014.** Performance comparison of genetic markers for high-throughput sequencing-based biodiversity assessment in complex communities. *Molecular Ecology Resources* **14**:1049–1059 DOI 10.1111/1755-0998.12254.