# Peer review of "Tackling critical parameters in metazoan meta-barcoding experiments: a preliminary study based on coxI DNA barcode"

_PeerJ, doi:10.7717/peerj.4845_

## Round 0.1 · original submission · Major Revisions

I have now received two reviews of your paper. Unfortunately, the reviewers were very split. I had a third reviewer lined up to help 'break the tie', but that reviewer has failed me after repeated attempts to get this third review after the reviewer committed. This also caused a delay in the decision process as I really wanted that third review. However, you've clearly waited long enough so I am cutting off that reviewer and moving forward with a Major Revision decision. As you can see, reviewer 2 especially has a number of important issues for your consideration. Note I will send a revised draft back to this reviewer for comment and hope that you can accommodate the reviewers' critiques in a reasonable way. Good luck with your revision.

Reviewer 1 ·

Basic reporting

Belech et al. developed a new pipeline HSTA (Hidden States Taxa Assign) for DNA meta-barconding.Moreover, the authors evaluated the influence of universal primers amplification bias and the correlation between GC content with taxonomic assignment accuracy.
Due to the importance that meta-barcoding is assuming in ecology, biodiversity description and conservation, these kind of study are very welcome.
I found the manuscript clear and well written. Title and abstract reflect the content. Introduction is exhaustive and methodology and results clearly described making the paper understandable also by research without particular bioinformatic skills.

Experimental design

The research fits with the aims of the journal.
The research question is clear and meaningful.
The methods used are well described

Validity of the findings

A comparative benchmark, carried out by using a “true” case study (community of terrestrial invertebrates) shows how the HSTA greatly improve the accuracy over other commonly used taxonomic assignment pipelines.
Lastly, the results highlight the necessity to evaluate the effect of GC content and universal primer set in any DNA meta-barcoding experiment.

Reviewer 2 ·

Basic reporting

Balech et al. investigate the impact of critical steps in the processing of amplicon data using the marker gene coxI which is commonly employed for meta-barcoding of animal populations. In particular, they have established a new approach for taxonomic classification and quantify their performance on mock community data and compare it with other established pipelines. Although the idea of using HMMs for classification of short sequencing reads is hardly a novel development, it is commendable that Balech and co-workers attempt to explore the effect of critical parameters in the assessment of taxonomic profiles using coxI, a question that has not been rigorously explored so far. I also found the manuscript to be well written and clearly accessible. I have however some concerns about the experiments performed that the authors should address, as well as relatively few remaining minor points.

Experimental design

1. As far as I understand, there is no replication in the mock community sequencing. This is clearly insufficient. At the very least technical replication is necessary to robustly assess whether the differences in performance are significant or comparable to the variability between replicates.

2. The authors build a custom database from sequences found in the actual samples and compare their method with de novo OTU picking algorithms. This is not entirely fair as it would be more appropriate to use for instance a reference-based algorithm as well such USEARCH-ref (Edgar 2013) with the same custom database (which would bypass the need of a taxonomic classifier such as RDP).

3. Similarly, the authors should compare multiple taxonomic classification algorithms and not only RDP.

Validity of the findings

4. An additional way to validate their approach would consist in retrieving published datasets and analyze them using the various pipelines. It would be relevant to see how large the resulting differences are and which methods consistently give similar results, etc. It would be possible to obtain OTU tables from various reference-based algorithms and for instance analyze their diversity using Bray-Curtis / UniFrac and PCoA, etc.

Additional comments

5. I find Fig. 1 somewhat confusing. For example, it is not clear to me if the Sanger sequences (LocaDB) are also included in the reference DB (CDS Reference Seq DB) used for the RDP classifier. I suggest the authors improve the current version for clarity and provide a more detailed description in the methods text or legend.

6. I found no link to the implementation of their approach. Since this study presents a computational method for analyzing data it seems desirable to be able to make it available for other groups to use besides necessary to ensure replicability of the results. Similarly, I did not find a link to a public repository where the data was deposited.

---

## Round 0.2 · Major Revisions

To summarise the situation with this submission:

After the previous Rejection decision, the authors contacted us to Appeal.

We provided the authors with the following feedback:

"There are really two issues here. One is the overgeneralization of the results. The authors indicate below they are focusing only on the relevant parameters, and not saying their methods is superior to other methods based on this single data set but in fact they state: “Our results showed that HSTA, a custom developed method, was more accurate and more constant to analyze coxI DNA barcode datasets compared to the other tested pipelines.” So, they could rewrite this manuscript to remove these kinds of statements, and that would be a big step in the right direction. But, the other part of the problem is the manuscript is poorly written to the point that you can’t be sure you would be correctly interpreting their intent while editing. This is what moved me to a reject."

After a couple of emails, the authors provided a new manuscript which appropriately addressed the language issues and 'toned down' the conclusions.

We are now at the point where the authors should upload the current version of their manuscript for a final decision to be made.

· Appeal

Appeal

Dear Professor Collins,

I am very sorry to hear that the manuscript has been rejected.

Although I have written the reasons for which we could not have a ‘classical’ replicate for our analyzed samples in the rebuttal letter, I would be happy to share with you some relevant points. Actually, the organisms we targeted in our analyses were already known, as they were characterized morphologically and their target coxI barcode region was sequenced singularly for each individual by Sanger method. When we massively sequenced the homogenized sample we were replicating the sequencing of the same individuals already sequenced singularly. In the manuscript, we have not mentioned unknown biodiversity, we were only limited to those organisms that we already know to be present in the samples. Moreover, we have validated the presence and the quantity of two organisms in qPCR, which was an additional confirmation of the results obtained from the analyses pipelines. We also reported the lack of resolution of HSTA at 3’ end of coxI barcode region and we justified it by analyzing the effect of GC content on sequencing quality.

Our aim in this manuscript was not absolutely the release of a new pipeline, which needs additional massive validations on a wide range of samples. It was only to highlight the relevant parameters worthy to consider while conducting similar experiments, namely GC content, organisms’ biomass, OTU picking threshold/s and the need of internal control sample to calibrate the analysis pipeline parameters. Probably, it would be convenient to change the title of the manuscript in order to stress only those important points as we have done in the discussions. Being accepted by one reviewer, I personally consider this study of relevant value for similar meta-barcoding experiments using coxI barcode region and I would be grateful if you can take into account these explanations and suggest re-considering the manuscript for publication even in another form such as short article.

Sincerely,

Bachir Balech.


· · Academic Editor

Reject

Dear Dr. Balech and Co-authors: I have been asked to handle this manuscript at a late stage as the previous Editor is unavailable.

I have read over the manuscript and the reviews. On the one hand, you have clearly put a significant amount of effort into a new approach which shows promise, and might be widely used if initial results are confirmed, and it is available is a usable format. On the other hand, I am forced to agree with the reviewer that comparisons of performance with this one rather idiosyncratic data set are insufficient to conclude that this method will generally be more accurate than other available methods. In order to draw that conclusion, comparison of a broader sampling of published datasets and or simulation studies with tests of significance would be required.

Reviewer 2 ·

Basic reporting

No comment

Experimental design

No comment

Validity of the findings

No comment

Additional comments

Unfortunately I have to persist in my previous comment concerning the lack of biological or technical replicates. Although I commend the effort the authors have put into this study I cannot recommend the manuscript for publication given this lack of replication.

---

## Round 0.3 · accepted · Accept

The language in the manuscript has improved tremendously, and the authors have done a nice job of toning down the earlier overdrawn conclusions. I think it is now ready to publish. I think the authors have done the best they can to address the issue of lack or replication and cast this as a preliminary, exploratory, rather than definitive study. On balance I think there is enough of significance to go ahead and put this out in the literature.

#